# Urogenital Schistosomiasis—History, Pathogenesis, and Bladder Cancer

**DOI:** 10.3390/jcm10020205

**Published:** 2021-01-08

**Authors:** Lúcio Lara Santos, Júlio Santos, Maria João Gouveia, Carina Bernardo, Carlos Lopes, Gabriel Rinaldi, Paul J. Brindley, José M. Correia da Costa

**Affiliations:** 1Experimental Pathology and Therapeutics, Research Centre, Portuguese Oncology Institute—Porto (IPO-Porto), 4200-072 Porto, Portugal; llarasantos@gmail.com (L.L.S.); lopes81241@gmail.com (C.L.); 2Department of Surgical Oncology, Portuguese Oncology Institute—Porto (IPO-Porto), Rua Dr. António Bernardino de Almeida, 4200-072 Porto, Portugal; 3Urology Department, Hospital Américo Boavida, Luanda 00200, Angola; juliosantos00@hotmail.com; 4Center for the Study in Animal Science (CECA/ICETA), University of Porto, Rua de D. Manuel II, Apt 55142, 4051-401 Porto, Portugal; mariajoaogouveia@gmail.com; 5Centre for Parasite Biology and Immunology, Department of Infectious Diseases, National Institute of Health Dr. Ricardo Jorge, Rua Alexandre Herculano 321, 4000-055 Porto, Portugal; 6Hormones and Cancer Lab, Institute of Biomedicine, iBiMED, Department of Medical Sciences, University of Aveiro, 3810-193 Aveiro, Portugal; carinadbernardo@gmail.com; 7Wellcome Sanger Institute, Wellcome Genome Campus, Hinxton, Cambridgeshire CB10 1SA, UK; gr10@sanger.ac.uk; 8Department of Microbiology, Immunology & Tropical Medicine, School of Medicine & Health Sciences, George Washington University, Washington, DC 20037, USA; pbrindley@email.gwu.edu; 9Research Center for Neglected Diseases of Poverty, School of Medicine & Health Sciences, George Washington University, Washington, DC 20037, USA

**Keywords:** schistosomiasis, squamous cell carcinoma, pathogenesis, bladder cancer

## Abstract

Schistosomiasis is the most important helminthiasis worldwide in terms of morbidity and mortality. Most of the infections occurs in Africa, which about two thirds are caused by *Schistosoma haematobium*. The infection with *S. haematobium* is considered carcinogenic leading to squamous cell carcinoma (SCC) and urothelial carcinoma of the urinary bladder. Additionally, it is responsible for female genital schistosomiasis leading to infertility and higher risk of human immunodeficiency virus (HIV) transmission. Remarkably, a recent outbreak in Corsica (France) drew attention to its potential re-mergence in Southern Europe. Thus far, little is known related to host-parasite interactions that trigger carcinogenesis. However, recent studies have opened new avenues to understand mechanisms on how the parasite infection can lead cancer and other associated pathologies. Here, we present a historical perspective of schistosomiasis, and review the infection-associated pathologies and studies on host–parasite interactions that unveil tentative mechanisms underlying schistosomiasis-associated carcinogenesis.

## 1. Introduction

Schistosomiasis or Bilharziosis is a neglected tropical disease (NTDs) caused by digenean parasites of genus *Schistosoma*. These metazoan parasites belong to the Phylum Platyhelminthes, Class Trematoda, and the genus *Schistosoma* includes five species infecting humans: *Schistosoma haematobium* [1], *Schistosoma mansoni* [2], *Schistosoma japonicum* [3], *Schistosoma intercalatum* [4], and *Schistosoma mekongi* [5]. The German pathologist Theodor Bilharz was the first to describe the parasite in 1851 [1]. In the course of post-mortem examinations carried out on Egyptian soldiers in Cairo, he described a putative parasite responsible for injuries in a letter to his mentor in Germany appointing them to the genus *Distomum*. In 1858, Weinland [6], remarking substantial and distinct differences between the novel parasite and the species of the genus *Distomum*, proposed a new genus: *Schistosoma* (from Greek schistos = split and soma = body). In honour to Theodor Bilharz, Cobbold in 1859 [7], proposed the amendment of the genus *Schistosoma* for Bilharzia. The name schistosomiasis prevails nowadays despite the frequent references to the disease as bilharziasis, in French and Portuguese medical literature. During the same year, Harley and Cobbold reported that the human infection occurred percutaneously [7,8]. Manson in 1902 [9], suggested the existence of two distinct species of *Schistosoma* and few years later, in 1908, schistosomiasis was detected for the first time in Brazil. The great contribution of Pirajá da Silva was the identification of two distinct species *S. haematobium* and *S. mansoni* [10]. The infections with these two species were finally established by Leiper in 1915 [11] who determined that the life cycle of the parasite included a freshwater snail as intermediate host. Concomitantly, in Asia, the rash of Kabure or the Katayama syndrome had been described in 1847, in a village near Hiroshima, in Japan. However, *S. japonicum* was only described in 1904, following the Katsurada incident [3]. Fujinamie and Nakamura in 1909 [12], studied the route of transmission of this species in the vertebrate host, exposing dogs and cats in the rice fields where human cases had been described. Subsequently, eggs were observed in faeces of those animals and when necropsied, 40 days after infection, adult worms were collected from the mesenteric veins of the portal system, confirming the presence of the parasite. *S. japonicum* has historically been described as the only zoonotic species of the genus *Schistosoma* [13]. However, recent findings have confirmed natural hybridization occurring in the human host among different schistosome species, including *S. haematobium*, *S. bovis,* and *S. curassoni*, with the latter two being agents of intestinal schistosomiasis in cows, sheep, and goats [14]. Moreover, evidence show that hybrid schistosomes are responsible for the recent schistosomiasis outbreak in Corsica [15].

During the Napoleonic invasion of Egypt, 1799–1801, French troops become infected by *S. haematobium*. French military physicians attributed the haematuria to the sweating and climate of Egypt, or as the “revenge of the Pharaohs”. During World War II, more than 1300 American soldiers become infected by *S. japonicum* during the invasion of Leyte in the Philippines. A few years later, during the preparatory military training for the invasion of Taiwan, executed in the Yangtze River basin, soldiers of the People’s Republic of China Red Army were exposed to *S. japonicum*. Many soldiers had developed Katayama fever, an occurrence that prompted the postponement of the planned amphibious assault on the island [16].

In the 1970s, Praziquantel (PZQ) was developed by Bayer [17,18], a pyrazino-isoquinoline derivative that displayed a powerful activity against parasitic flatworms, including schistosomes. Even though it has been more than 40 years since its development, PZQ continues to be the key drug against schistosomiasis. During the first decades of the 21st century, mass drug administration (MDA) programmes against schistosomiasis that rely on PZQ were launched in endemic regions in Africa, Latin America, and the Middle East [19,20]. For instance, in 2018 alone, ~235 million people received PZQ tablets [21]. Consequently, the global disease burden has dropped. However, the long-term efficacy for optimal morbidity control or transmission elimination by MDA approaches is still controversial, in particular in regions where people live under deficient or non-existent sanitation conditions and are constantly exposed to the parasite. Moreover, reliance on only a few drugs, and their mass-administration is likely to drive the development of drug resistance [22]. Nowadays, more than 90% of the cases of schistosomiasis occur in Africa, two-thirds of which are caused by *Schistosoma haematobium* [23,24,25]. Indeed, the actual number of cases of urogenital schistosomiasis (UGS) may far exceed that previously predicted. Moreover, female genital schistosomiasis increases the risk of human immunodeficiency virus (HIV) transmission [26,27,28] and a recent outbreak of urogenital schistosomiasis in Corsica confirms its re-emergence in Europe [29,30]. In addition to directly damaging development, health and prosperity of infected populations, the chronic infection with *S. haematobium* eventually leads to squamous cell carcinoma (SCC) of the urinary bladder [31]. Thus far, little is known about the interaction host-parasite that triggers carcinogenesis.

In 2009 schistosome species moved into the post-genomic era, as the first draft genomes for *S. mansoni* and *S. japonicum* were published [32,33], and a few years later the *S. haematobium* genome was released [34]. In the last decade, a remarkable functional technology development has occurred, from culture system refinements [35,36], the use of functional tools, such as transgenesis and genome editing [37,38,39,40], to single cell transcriptomics [41], has accelerated the discovery of targets for novel control strategies

## 2. Urogenital Schistosomiasis: Pathogenesis and Cancer

Urogenital schistosomiasis (UGS) is caused by egg-laying *S. haematobium* worms dwelling within the veins draining the main pelvic organs, including the bladder, uterus, and cervix. The infectious stage of parasite, larvae cercariae that emerge from freshwater snails infect humans through direct skin penetration. The worms migrate into the circulation, mature and lodge within the venous plexus of the bladder, where they reproduce, and females release up to 3000 eggs per day. Only half of these eggs are excreted in the urine to propagate the parasite’s life cycle, whereas the remaining eggs become entrapped within capillary beds of the pelvic end organs, especially in the bladder, ureters, and genital tract. UGS is characterized by chronic immune mediated disease. The continuous inflammatory reaction to the eggs leads to parenchymal tissue destruction, inflammation, fibrosis, granulomata, and ultimately to fibrotic nodules termed sandy patches (Figure 1) [42].

The lesions induced by the entrapped eggs lead to bladder and ureteral inflammation and hematuria in >50% of cases, in addition to organ deformities such as narrowing of the ureters. Secondary urinary tract and renal infections, hydronephrosis, and ultimately renal failure were observed in millions of people [42]. Importantly, bladder cancer is a frequent and dire complication of chronic UGS. The incidence of SCC associated to UGS is estimated in 3–4 cases per 100,000 [43]. Untreated patients often develop schistosome-related bladder cancer. The tumors are found in a relatively younger age group, commonly present as well differentiated SCC locally advanced and have poor overall survival [44,45]. The severity and frequency of UGS sequelae are related to the intensity and duration of the infection [46,47,48]. Genetic alterations, chromosomal aberrations, and cytological changes have been described in carcinomas associated with UGS [42,46,49]. *N*-nitroso compounds have been implicated as tentative etiologic agents in the process of bladder carcinogenesis [50]. Elevated levels of DNA alkylation damage in carcinomas associated with UGS and a high frequency of G to A transitions in the H-ras gene and in the CpG sequences of the p53 tumor suppressor gene have also been reported [46,48]. These outcomes indicate that UGS-associated SCC arises through a progressive accumulation of genetic changes in epithelial cells. Positive correlation between UGS and increased the levels of oxidative stress accompanied by continuous DNA damage and repair in urothelial carcinomas has been observed by several groups [51,52,53]. More recently, we have shown that schistosome eggs co-cultured with informative human cell lines promote proliferation of the urothelial cells (HCV29 cell line) but inhibit cholangiocytes (H69 cells). Moreover, the TP53 pathway was significantly downregulated, and the estrogen receptor was predicted to be downregulated in urothelial cells exposed only to *S. haematobium* but not *S. mansoni* eggs [54].

## 3. Evidence of Chemical Carcinogenesis as Initiator of Bladder Carcinogenesis Associated with Urogenital Schistosomiasis

An oestrogen-DNA adduct mediated pathway in UGS-associated bladder cancer has been postulated [52,55,56,57]. *S. haematobium* eggs and adult worm lysates stimulate cellular proliferation, interfere with apoptosis, increase oxidative stress, and induce a genotoxicity on HCV-29 cells, derived from urothelial cells [52,58,59]. In addition, we have identified and characterized oestrogen-like metabolites in the lysates and secretions of *S. haematobium* worms and eggs [58], and in sera and urine of UGS cases [52,56,58]. Remarkably, the downregulation of the oestrogen receptor predicted to occur in urothelial cells exposed to live eggs seem to have been species-specific, i.e., only *S. haematobium* eggs induced this effect in cells, but not the *S. mansoni* eggs [54].

Inspired by the chemical carcinogenesis phenomenon described for several types of cancer [60], we have hypothesized that reactive metabolites derived from schistosomes might be involved in the SCC carcinogenesis associated to UGS [55,57]. Hydroxylation of estrogens forms the 2- and 4-catechol estrogens involved in further oxidation to semiquinones and quinones, including the formation of the catechol estrogen-3,4-quinones, and the major carcinogenic metabolite of estrogen have been reported [52,56]. These electrophilic compounds can react covalently with macromolecules including DNA to form the depurinating adducts that eventually generate mutations in proto-oncogenes and/ or tumor suppressors consequently leading to carcinogenic progression [55,57]. We also have reported alterations in *p53* in most schistosome-associated bladder tumors, irrespectively of their histopathological nature [61]. In addition, the Ki-67 overexpression was less evident in pre-malignant lessons compared to its overexpression in already established urothelial and squamous cell carcinomas [61]. Estrogen receptors mediate cell proliferation, increasing errors during the DNA replication [62,63].

Oestrogen metabolite species were detected as main constituents of urine from different groups of individuals with UGS [56]. The mass spectrometric profile of urine samples of 40 patients with UGS revealed the presence of seven specific metabolites. Nevertheless, these metabolites were more pronounced in individuals without non-malignant lesions compared to cases with UGS-related cancer [56]. This evidence suggests that DNA oxidation is more evident in former cases that the latter ones. Could this be an indicative of the initiation/progression of carcinogenesis? The study of these molecules not only informs about the basic pathophysiology of the cancer associated with the infection, but also offers translational avenues such as the identification of the molecules as tentative biomarkers. The molecules derived from 8-oxodG or estrogen derivatives (Figure 2) might be traced as urine biomarkers for the early detection and progression of bladder cancer [56]. 8-oxodG is a representative marker for DNA oxidative damage during oxidative stress [64,65,66], the detection of 8-oxodG in urine samples suggests that DNA damage occurs during UGS. These reactive oxygen species induce oxidation, nitration, halogenations, and deamination of biomolecules, including nucleic acids, with the formation of toxic and mutagenic products [51]. It is important to note that this study has some limitations, such as a lower number of samples and the comparison with a database of healthy individuals [66] instead of individuals from endemic areas. Further investigations are warranted to validate this evidence.

## 4. Urine Proteome in Urogenital Schistosomiasis

The urine proteome profiling of patients with UGS alone, UGS-related bladder cancer, and bladder cancer without infection highlighted the activation of distinct molecular pathways. Purified urine proteins separated by sodium dodecyl sulfate-precast polyacrylamide gel electrophoresis (SDS-PAGE) and analysed by mass spectrometric (GeLC-MS/MS) followed by protein-protein interaction analysis revealed Th2-type immune response and oxidative stress as the most prevalent biological processes in UGS samples. On the other hand, UGS-related bladder cancer was associated with proteins involved in inflammation and negative regulation of endopeptidase activity whereas in the case of bladder cancer without infection, proteins were mainly associated with metabolism, cell adhesion, tumor growth and metastasis, and immune response. This study showed the role of Th2-type immune response and chronic inflammation as major drivers of schistosome induced carcinogenesis and revealed a set of proteins that should be further explored in a multimarker strategy for the early diagnosis of schistosome-related bladder cancer [67].

The *Schistosoma japonicum* genome, revealed shared sequences with humans for mammalian-like receptors for insulin, progesterone, cytokines and neuropeptides, suggesting that host hormones or endogenous parasites analogues might coordinate parasite development and maturation and that *Schistosoma* modulates host immune responses through inhibitors, molecular mimicry and other invasion strategies [68]. Genome sequencing and analysis of *S. haematobium* proteome also showed molecules linked to immunomodulation such as inhibition of antigen processing and Th2 responses [34,69]. Similarly, quantitative photocytometry analysis of tissue samples from UGS-related bladder cancer using T cells specific antibodies showed an unbalanced Th1/Th2 relation in which Th2 was upregulated and dominated [70]. Both inflammation and alternative complement pathway activation are upregulated in this situation. A prolonged inflammatory response might lead to increased DNA mutation rates, due to previous alterations in suppressor genes such as TP53 (expressed by accumulation of p53 in the urothelium) and overall genetic instability, characterized by high levels of 8-nitroguanine and 8-hydroxy-2′-deoxyguanosine (8-oxodG) [56,60,71].

The consonance of data obtained in the proteomic and spectrometric studies supports the hypothesis that most cancers may have origin from a biological and/or chemical insult that trigger a sequence of events that culminate with formation of a pre-cancerous niche [72] better understanding of the mechanisms underlying immunomodulation during chronic schistosomiasis and UGS-related carcinogenesis will provide new insights to optimize treatment strategies of schistosomiasis and prevention or treatment of UGS-related bladder cancer.

## 5. Alterations in the p53 Pathway Associated with UGS-Associated Bladder Cancer

Loss of p53 function leads to alteration of cell proliferation, cell longevity and resistance to cytotoxic drugs [73]. The functional changes of this gene suppressor (TP53), broadly, mutations that besides reducing the concentration of MDM2 (i.e., a transcriptional target of p53) favour the transcription and protein phosphorylation of p53 and form the basis of the loss of capacity. Normal p53 is degraded quickly after its synthesis; the same is not checked in the form dictates, as amended, the mutated p53 that features an increased half-life, favouring its intracellular accumulation. The expression of p53 in urothelial carcinomas is intense and involves both cell clones of malignant cells as well as adjacent, apparently normal urothelial cells of the mucosa [61].

What is the role of p53 in UGS-associated bladder cancer? The studies covering this issue showed that the tumor suppressor protein p53 was overexpressed in bladder urothelial mucosa in individuals with UGS or SCC-associated to UGS. What is the meaning of this finding? First, it is essential to compare these findings with the most frequent pattern of expression of p53 in normal bladder urothelium. p53 is expressed between 1 and 37% of cases in only 5% of urothelial cells and scattered. In synthesis, the expression of this protein in physiological situations is infrequent and transitory [74,75]. TP53 is not only overexpressed in bladder urothelial mucosa of individuals with UGS-bladder cancer but also in those with only UGS; thus, could this be considered as a biomarker of malignant transformation of the bladder tissue associated with the *S. haematobium* infection?

The expression of p53 in cells of the urothelium and the occurrence of mutations of TP53 gene in comparative studies suggest that the profuse expression of p53 is associated with mutations in the TP53 gene [73,76,77,78,79,80,81,82]. The expression of p53 was studied in malignancies associated or not to *S. haematobium* infection in urothelial and squamous cells. The expression of p53 was significantly higher in little differentiated tumors however, the expression of p53 prevailed in locally advanced tumors in UGS [61].

Badawi et al. [82] found that patients with UGS, presented a high number of DNA lesions associated with the action of alkylating agents. The mutations occurred simultaneously with disruption of the DNA repair mechanisms; more, described that these mutations were caused by G-transitions the H-ras gene and CpG sequences on the TP53 gene. Finally, Abdulamir et al. [83] found that p53 expression was more frequent in malignant tumors associated with UGS than other tumors of the bladder. Also, Kamel et al. [84] referred that any clinical case of squamous metaplasia and hyperplasia in bladders, not associated with UGS, has increased expression of p53. Our results support these earlier findings. Increased p53 expression was detected in cases with cystitis without tumor, in cases with malignant neoplasms, squamous cell, and urothelial or mixed carcinoma, and in cases with apparently normal mucosa adjacent to the tumor. The expression of p53 in UGS patients, either with associated bladder cancer or not, was high, involving a large number of adjoining cells, suggesting accumulation of p53 at nuclear level [61]. This pathophysiological event occurs when the p53 gene was mutated and there is no degradation of the protein. In these circumstances the role of p53 is disturbed, which allows the accumulation of changes in DNA and its transmission to daughter cells. In the absence of repair and apoptosis, new mutations may follow, an environment conducive for malignant transformation. In in vitro experiments we showed a downregulation of the p53 pathway in urothelial cells exposed to either live *S. haematobium* or *S. mansoni* eggs. However, the genes that led to this overall downregulation of the pathway were different for each of the two schistosome species [54]. In synthesis, the concatenation of all these molecular events in the UGS patients support a relevant role to *S. haematobium* in initiation of bladder carcinogenesis, may be mediated by oestrogen metabolites, followed by chronic inflammation, fibrosis, and changes in the tissue microenvironment including mutations in key tumor suppressor genes and proto-oncogenes. In addition, strong evidence based on animal models of *Schistosoma haematobium* infection support the role of p53 in the development of UGS-associated SCC. The animal model of the infectious disease developed by Michael Hiseh is based on the injection of *S. haematobium* eggs directly to the bladder of the mouse. This led to urinary tract fibrosis, bladder dysfunction and other alterations of morphology consistent with human UGS [85,86]. Using this model in transgenic mice, Honeycutt and colleagues [87] found that alteration in the p53 signalling in the urothelium might affect the normal tissue homeostasis during UGS.

## 6. Biomarker Candidates for UGS and SCC of the Bladder?

Biomarkers for early detection and prognosis of malignancy induced by UGS are needed. Promising candidates, notably (1) oestrogen-like and (2) 8-oxodG related metabolites highlighted herein, appear worthy of validation in larger population-based studies. In addition, we caution that the malignant lesions included here included both UCC (urothelial cell carcinoma) and SCC. Mixed urothelial cancer with squamous features, such as some cases in individuals with UGS and bladder cancer (either SCC or urothelial cancer cell) described in [56], may display a distinct pathophysiology compared to UGS related ‘pure’ SCC, and the literature clearly indicates that UGS is a risk factor for SCC but not UCC [48,52]. Accordingly, investigations of larger populations may also facilitate the identification of metabolites that characterize the discrete pathogenesis of SCC and UCC.

## 7. Conclusions

Despite all the efforts schistosomiasis remain a great concern for public health in developing countries. *S. haematobium* is considered a carcinogenic biological agent and bladder cancer is one of the direst complications affecting individuals from young age. What should be the focus of research on schistosomiasis? We believe that the research should focus on three main components: (1) understanding the host–parasite interactions (e.g., immune response elicited by infection); (2) mechanisms underlying the UGS-related carcinogenesis; and (3) novel control strategies for both schistosomiasis and associated carcinogenesis. Further studies in vitro and in vivo using animal models will shine a light on these components.

## Figures and Tables

**Figure 1 jcm-10-00205-f001:**
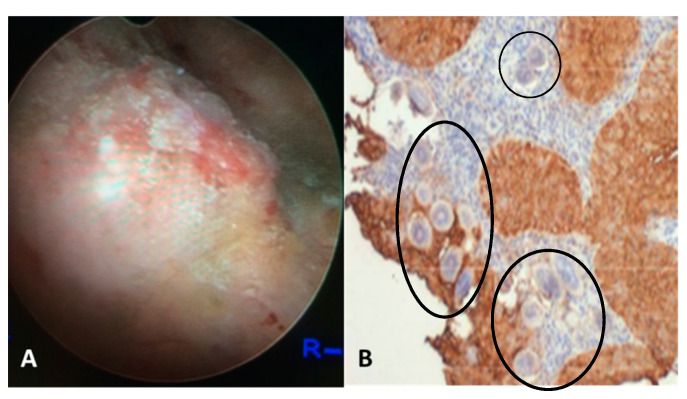
(**A**) Cystoscopy view of the posterior bladder mucosa with UGS lesions, granulomas, ulcers and tumor. (**B**) Histology from the bladder mass biopsy showing *S. haematobium* ova (black circles) and SCC of the bladder (expression of sialyl-Le^a^).

**Figure 2 jcm-10-00205-f002:**
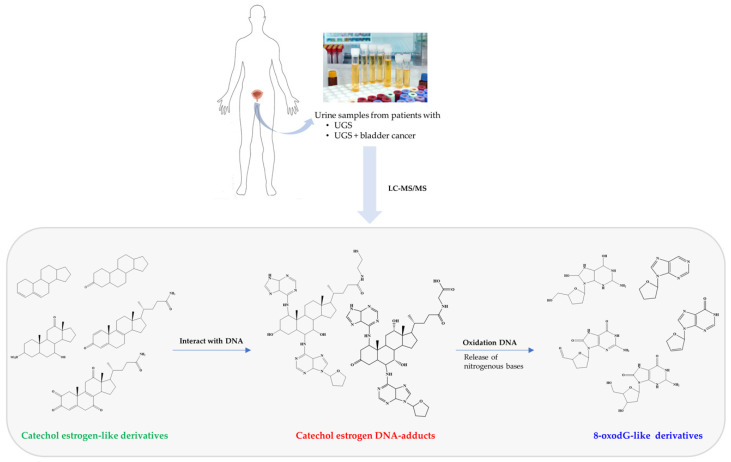
Metabolites identified in urine with UGS and UGS-associated SCC that were not present in urine of healthy individuals [65]. The metabolites were divided in three groups: catechol estrogen-like (**green**), catechol-estrogen-DNA adducts (**red**) and 8-oxodG derivatives (**blue**). The catechol estrogen-DNA adducts may be a consequence of interaction of catechol estrogen derivatives with host DNA. On other hand, 8-oxodG derivatives may be a result of liberation of nitrogenous bases from DNA and/or its oxidation.

## Data Availability

The data presented in this study are available in references [52,56,58,59,61,67].

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
