# Peer review of "Urogenital Schistosomiasis—History, Pathogenesis, and Bladder Cancer"

_jcm, 2021, doi:10.3390/jcm10020205_

Round 1
Reviewer 1 Report
I am surprised the authors have not cited the extensive literature on the mouse model of urogenital schistosomiasis (by Michael Shieh’s group), much of which is focused on pathology and carcinogenesis. Before this corpus there was no description of the earliest pathologic changes of UGS, due to not knowing exactly when patients were infected and the inability to ethically obtain bladder tissue from these patients. The manuscript is incomplete without a review of the associated papers. The authors themselves point out in their Conclusion that "research should focus …. understanding the host-parasite interactions (e.g. immune response elicited by infection)”. Sheih’s work has addressed some of these topics.
Minor issues
The abstract has typographical errors: “interactions host-parasite” and “unvail” (should be “unveil”)
Typographical errors: Line 50 “bilharziosis” should be “bilharziasis”
Line 80: Roman numerals to denote the 21st century are not appropriate
Line 83: “tables” should be “tablets”
Line 96: “schistosomes species”
Lines 104-105 “parasite infectious larvae cercariae”
Figure 1. Black arrows are not ideally positioned to indicate eggs
Lines 119-120: "Secondary urinary tract and renal infections, hydronephrosis, and ultimately renal failure in millions of people” is not a complete sentence.
Line 175 "reduce number of samples”
Line 213 "[6273A”
Line 221: "protein phosphorylation p53"
Line 226 "UGS-associate"
Author Response
I am surprised the authors have not cited the extensive literature on the mouse model of urogenital schistosomiasis (by Michael Shieh’s group), much of which is focused on pathology and carcinogenesis. Before this corpus there was no description of the earliest pathologic changes of UGS, due to not knowing exactly when patients were infected and the inability to ethically obtain bladder tissue from these patients. The manuscript is incomplete without a review of the associated papers. The authors themselves point out in their Conclusion that "research should focus …. understanding the host-parasite interactions (e.g. immune response elicited by infection)”. Sheih’s work has addressed some of these topics.
We acknowledge the reviewer for bringing this up. We agree the contributions by the Hsieh lab have been key in this field. Therefore, we have now incorporated and discussed some of his critical contributions. In this regard, please see the new paragraph (lines 266-271)
Minor issues
The abstract has typographical errors: “interactions host-parasite” and “unvail” (should be “unveil”)
We corrected this. Please see line 34.
Typographical errors: Line 50 “bilharziosis” should be “bilharziasis”
We corrected this. Please see line 50.
Line 80: Roman numerals to denote the 21st century are not appropriate
We corrected this to 21st century.
Line 83: “tables” should be “tablets
We corrected this. Please see line 83.
Line 96: “schistosomes species”
We corrected to schistosome species.
Lines 104-105 “parasite infectious larvae cercariae”
We corrected to “..the infectious stage of parasite, larvae cercariae..”
Figure 1. Black arrows are not ideally positioned to indicate eggs
We replaced the arrows with circles.
Lines 119-120: "Secondary urinary tract and renal infections, hydronephrosis, and ultimately renal failure in millions of people” is not a complete sentence.
We corrected the sentence….”Secondary urinary tract and renal infections, hydronephrosis, and ultimately renal failure in were observed in millions of people.”
Line 175 "reduce number of samples”
We replaced “reduce number of samples” by “..lower number of samples”
Line 213 "[6273A”
It is a typo. We corrected to “…[73]..”.
Line 221: "protein phosphorylation p53"
We corrected to “..protein phosphorylation of p53…”
Line 226 "UGS-associate"
We corrected to “..UGS-associated…”
Reviewer 2 Report
This is a relevant overview of literature in the field of urogenital schistosomiasis. Although it does not cover classical immunological topics, it does discuss host-pathogen interactions on a cellular and molecular level.
I have a few small comments:
-Although epidemiology is discussed, I miss the actual numbers on UGS carcinogenic complications. Could the authors provide the incidence of SCC and/or UCC, or if not available, at least comment on this knowledge gap?
-I noticed that animal models are not included here. Have animal studies not been published on this topic?
-Many data are discussed on genetic and metabolic markers of UGS-associated malignancies. All these studies compared different cell lines or patient groups, or sometimes did not have an appropriate control. It would be useful to add some ideas on a future strategy for further investigation into this topic (e.g. animal models? better designed studies?).
Author Response
This is a relevant overview of literature in the field of urogenital schistosomiasis. Although it does not cover classical immunological topics, it does discuss host-pathogen interactions on a cellular and molecular level.
I have a few small comments:
-Although epidemiology is discussed, I miss the actual numbers on UGS carcinogenic complications. Could the authors provide the incidence of SCC and/or UCC, or if not available, at least comment on this knowledge gap?
In our point of view, the scope of the present manuscript is not related with epidemiological data. Anyway, in 2014, we have published a review article that described this data (Costa et al., Front Genet 2014, 5:444). Please see these two citations “Of ∼112 million cases of S. haematobium infection in sub-Saharan Africa, 70 million are associated with hematuria,18 million with major bladder wall pathology, and 10 million with hydronephrosis leading to kidney damage (van derWerfetal.,2003; Hotezetal.,2009; King, 2010).” and “The incidence of urogenital schistosomiasis associated SCC is estimated in 3–4 cases per 100,000 (Shiff et al., 2006).“ However, we have included in the manuscript data related with the incidence of SCC associated with UGS in line 121.
-I noticed that animal models are not included here. Have animal studies not been published on this topic?
The response is related to the first point of reviewer 1 (see above). We have now incorporated information on the use of animal models to understand the process of bladder carcinogenesis associated with the infection with Schistosoma haematobium. Please see new lines 266-271.
-Many data are discussed on genetic and metabolic markers of UGS-associated malignancies. All these studies compared different cell lines or patient groups, or sometimes did not have an appropriate control. It would be useful to add some ideas on a future strategy for further investigation into this topic (e.g. animal models? better designed studies?).
Please see the new sentence in line 279.
Round 2
Reviewer 1 Report
The authors have addressed my concerns